# Chemical Profile Analysis of *Prosopis laevigata* Extracts and Their Topical Anti-Inflammatory and Antibacterial Activities

**DOI:** 10.3390/plants14071118

**Published:** 2025-04-03

**Authors:** Manasés González-Cortazar, David Osvaldo Salinas-Sánchez, Maribel Herrera-Ruiz, Paulina Hernández-Hernández, Alejandro Zamilpa, Enrique Jiménez-Ferrer, Beatriz E. Utrera-Hernández, Ma. Dolores Pérez-García, Ana S. Gutiérrez-Roman, Ever A. Ble-González

**Affiliations:** 1Centro de Investigación Biomédica del Sur, Instituto Mexicano del Seguro Social, Argentina No. 1, Col. Centro, Xochitepec 62790, Morelos, Mexicoazamilpa_2000@yahoo.com.mx (A.Z.); enriqueferrer_mx@yahoo.com (E.J.-F.); lola_as@yahoo.com.mx (M.D.P.-G.); 12gtr.ana@gmail.com (A.S.G.-R.); 2Centro de Investigación en Biodiversidad y Conservación (CIByC), Universidad Autónoma del Estado de Morelos (UAEM), Av. Universidad 1001, Col. Chamilpa, Cuernavaca 62209, Morelos, Mexico; 3Escuela de Estudios Superiores del Jicarero (EESJ), Universidad Autónoma del Estado de Morelos, Carretera Galeana-Tequesquitengo s/n Col. el Jicarero, Jojutla 62909, Morelos, Mexico; 4División Académica de Ciencias Básicas, Universidad Juárez Autónoma de Tabasco, Carretera Cunduacán-Jalpa Km. 0.5, Cunduacán 86690, Tabasco, Mexicoever.ble@ujat.mx (E.A.B.-G.); 5Centro de Desarrollo de Productos Bióticos, Instituto Politécnico Nacional, Carretera Yautepec-Jojutla s/n km 85, San Isidro 62739, Morelos, Mexico

**Keywords:** *Prosopis laevigata*, inflammation, antibacterial, cytokines, ethyl veratrate

## Abstract

There are two major global morbidity and mortality problems in the health sector: inflammation, which is the physiological process that, in acute and chronic conditions, gradually causes the loss of the body’s functionality, leading to severe damage to health; and microbial diseases, which are caused by pathogenic microorganisms such as bacteria. In the present study, the anti-inflammatory effects of three extracts of mesquite (*Prosopis laevigata)*—*n*-hexane (PH), dichloromethane (PD), and methanol (PM)—were assessed in a mouse model of 12-*O*-tetradecanoylphorbol-13-acetate (TPA)-induced ear oedema, and the antimicrobial effects against 14 microorganisms were assessed using the broth microdilution method. The extracts inhibited ear oedema by 60.81% (PH), 75.96% (PD), and 60.29% (PM). The most active anti-inflammatory extract (PD) was fractionated through chromatography, and three fractions (PDR3, PDR6, and PDR7) were evaluated. One of the most active fractions (PDR7) was purified via column chromatography, and ethyl veratrate (VE, **1**) was isolated and identified. VE inhibited ear oedema by 85.1%. The anti-inflammatory effect is evidenced by the quantification of two pro-inflammatory cytokines (IL-10 and TNF-α). The PD extract, the PDR7 fraction, and the compound present an IL-10 concentration of 11.8, 18.9, and 36.5 pg/mg of protein, values significantly higher than the group that received only phorbol ester (* *p* < 0.05). These treatments also significantly decreased the concentration of TNF-α (* *p* < 0.05) to 197.6, 241.9, and 247.0 pg/mg protein, respectively. The PM extract showed the most pronounced antimicrobial effect, with a minimum inhibitory concentration (MIC) of <12.5 µg/mL for almost all the 14 tested strains, followed by the PD and PH extracts. Chromatographic fractionation of the PM extract yielded the PMR6, PMR7, and PMR10 fractions that inhibited all tested microorganisms with a MIC between 6.25 and 200 µg/mL. Compound **1** was active on five strains, with a concentration between 2 and 8 µg/mL. High-performance liquid chromatography analysis and comparison with commercial standards allowed for the identification of rutin (**2**) and quercetin 3-*O*-glucoside (**3**). Gas chromatography–mass spectrometry analysis of the PH and PD extracts allowed for the identification of fatty acids, terpenes, and phenols.

## 1. Introduction

Antimicrobial resistance is a very serious global public health problem and represents an urgent threat due to the presence of resistance to antimicrobial agents [1]. These resistant microorganisms—including bacteria, fungi, and parasites, among others—cause infections and generate great morbidity and mortality throughout the world [2]. According to recent reports, it is estimated that by 2050, there will be 10,000 deaths each year due to this public health problem [3]. Antimicrobial agents have several mechanisms of action, including the inhibition of cell wall synthesis, protein synthesis, nucleic acid synthesis, and metabolic pathways, as well as the depolarisation of the cell membrane [4]. Resistance can result from numerous phenomena, including the modification of targets, where the microorganism acquires a mutation that alters the drug binding site; the production of inactivating enzymes (e.g., beta-lactamase, which breaks down the beta-lactam ring of penicillin); efflux pumps in the membranes that expel drugs (e.g., tetracycline); membrane impermeability (e.g., the outer membrane in Gram-negative bacteria); and alternative metabolic pathways that utilise enzymes that are not inhibited by the available antimicrobial agents [4,5,6]. After a microorganism causes an infection, inflammation is one of the responses to the pathogen [7,8]. Inflammation is a natural defense response of the body’s cells and tissues against endogenous or exogenous agents (e.g., bacterial and fungal infections, trauma, necrosis, chronic degenerative diseases, and chemical agents).

Traditional medicine using medicinal plants plays a crucial role in the treatment of diseases, including microbial infections and excessive inflammation. Indeed, extracts from these plants may help to enhance or complement medical treatments for inflammatory diseases while preventing antimicrobial resistance [9,10]. Many plant species synthesise natural products with direct antimicrobial activity, among which flavonoids, terpenes, and alkaloids stand out. For example, garlic (*Allium sativum*) has shown efficacy against multidrug-resistant bacteria [11]. Green tea (*Camellia sinensis*), which has a high phenolic compound content, can interfere with bacterial efflux pumps, which are responsible for expelling antibiotics from the cell [12]. Likewise, secondary metabolites isolated from plants, such as rosmarinic acid, enhance the effects of antibiotics against resistant bacteria [13]. In addition, some plants are used to suppress inflammation, whether chronic or acute. The mechanism of action is quite variable. For example, gingerol in ginger (*Zingiber officinale*) inhibits the release of proinflammatory cytokines such as interleukin 1 (IL-1), IL-6, and tumour necrosis factor alpha (TNF-α) and activates nuclear factor kappa B (NF-κB), which regulates the inflammatory response [14].

The *Prosopis* genus comprises more than 44 species distributed mainly in Africa, America, and Asia. These species are generally small trees or shrubs. Herbal preparations obtained from *Prosopis* species have various ethnomedicinal uses, including to treat asthma, rheumatism, poisonous bites, pests, and plant diseases. The compounds that have been isolated and identified include phenolics, alkaloids, flavonoids, and terpenes [15]. Their pharmacological properties include antibacterial, antifungal, anti-inflammatory, antitumor, and analgesic activities [16,17].

*Prosopis laevigata*, commonly called smooth mesquite, is a small tree or shrub widely distributed throughout Mexico. In traditional Mexican medicine, it is used to treat eye diseases, rash, and digestive system disorders [18]. *P. laevigata* leaves have also been reported to have antimicrobial activity [19,20], anthelmintic activity [21], and antioxidant and cardioprotection potential [19]. Two alkaloids have been isolated and identified: prosopine and prosopinine [22,23]. In a recent study, researchers identified a flavonoid known as isorhamnetin from the ethyl acetate fraction; it prevented nematicidal activity against the eggs and larvae of *Haemonchus contortus* [21]. This work aimed to determine the chemical composition of three *P. laevigata* extracts—*n*-hexane (PH), dichloromethane (PD), and methanol (PM)—as well as their anti-inflammatory properties in a mouse model of 12-*O*-tetradecanoylphorbol-13-acetate (TPA)-induced ear oedema and as an antibacterial against 14 microorganisms.

## 2. Results

### 2.1. High-Performance Liquid Chromatography (HPLC) Analysis of the PM Extract

The HPLC chromatograms (Figure 1) show the chemical profile of the PM extract, as well as the rutin (**2**) and quercetin 3-O-glucoside (**3**) standards. A chromatographic comparison of the retention times and ultraviolet (UV, see in Appendix A) spectra with the standards indicated that the peaks at 8.81 min (λ_max_ = 212.2, 256.9.7, and 355.3 nm) correspond to compound **2** and 9.11 min (λ_max_ = 212.2, 255.7, and 355.3 nm) to compound **3** (see Figure 2).

### 2.2. The Gas Chromatography–Mass Spectrometry (GS-MS) Profiles of the PH and PD Extracts

The phytoconstituents present in the PH extract are shown in Table 1. The chemicals include alkanes (1,5,4-dibromotetrapentacontane, tetracontane, heptacosane, and tetratetracontane), a fatty acid (hexanedioic acid), an alkylated phenol (6,6′-di-tert-butyl-4,4′-diethyl-2,2′-methylenediphenol), and a terpene (squalene).

Table 2 presents the chemical constituents of the PD extract. They include a benzofuran (6-hydroxy-4,4,7a-trimethyl-5,6,7,7a-tetrahydrobenzofuran-2(4H)-one), a diterpene alcohol (phytol), and a fatty acid (bis[(2S)-2-ethylhexyl] hexanedioate).

### 2.3. Anti-Inflammatory Effect of the P. laevigata Extracts

Table 3 shows the anti-inflammatory activity of the PH, PD, and PM extracts based on the mouse TPA-induced ear oedema model (applying 1 mg/ear). The PD extract had the greatest anti-inflammatory effect; it inhibited oedema by 75.96% (2.74 mg), higher than the positive control (indomethacin: 65.08% and 4 mg). The PH and PM extracts showed 60.81% (4.00 mg) and 60.29% (4.54 mg) inhibition, respectively.

#### Structural Elucidation of Compound (**1**)

Compound (**1**) was isolated as a yellow powder. In the UV spectrum, the compound showed a λ_max_ of 211, 249.8, 307.7, and 347 nm, characteristic of a phenolic compound. The ^13^C nuclear magnetic resonance (NMR) spectrum of **1** (Table 4, see figure in Appendix A) showed 11 peaks, of which 4 are quaternary, 3 are methines, 1 is a methylene, and 3 are methyls. The ^1^H NMR spectrum showed an ABX aromatic ring system at δ 7.58 (1H, d, J = 2.0 Hz, H-3), 6.77 (^1^H, d, J = 8.9 Hz H-6), and 7.57 (1H, dd, J = 2.0 and 8.4 Hz, H-7). Additionally, there were signals at δ 3.83 (s) and 3.82 (s), which, according to HSQC, correspond to two methoxyls at δ 55.8; and the presence of an ethyl acetate at δ 4.25 (2H, q, 7.5) and 1.29 (3H, t, 6.8), which are assigned to H-1′ and H-2′, respectively. According to the analysis of the spectroscopic data (Table 3), it is proposed that **1** is a tri-substituted aromatic ring derived from veratric acid, specifically ethyl veratrate (VE, **1**) (see Figure 2).

**Table 4 plants-14-01118-t004:** The proton (^1^H) and carbon (^13^C-NMR) nuclear resonance spectroscopy data for compound **1** (CDCl_3_, 600 MHz).

Position	δ^1^H (δ in ppm, J in Hz) 1	δ^13^C (HSQC) 1	HMBC (J_2–3_)
1		166.2	7.57, 7.45, 6.77, 4.26
2		122.9	
3	7.45 (1H, d, 2.0)	110.1	123.3, 148.5, 152.8, 166.5
4		148.5	
5		152.8	
6	6.77 (1H, d, 8.9)	111.8	122.9, 148.5, 152.8
7	7.57 (1H, dd, 2.0, 8.9)	123.3	111.8, 148.5, 152.8, 166.2
1′	4.25 (2H, q, 7.5)	60.6	14.2, 166.2
2′	1.29 (3H, t, 6.8)	14.2	60.6
OCH_3_	3.83, s	55.8	152.8
OCH_3_	3.82, s	55.8	148.5

**Figure 2 plants-14-01118-f002:**
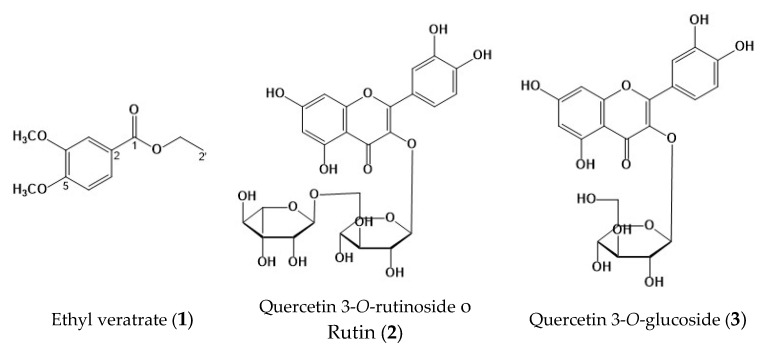
Chemical structures identified in *Prosopis laevigata*: ethyl veratrate (**1**); quercetin 3-*O*-rutinoside o rutin (**2**); quercetin 3-*O*-glucoside (**3**).

### 2.4. Anti-Inflammatory Effect of PD Extract Fractions and Compound (**1**)

The PD extract was separated into eight fractions to identify the active principles responsible for the anti-inflammatory activity. The PDR3, PDR6, and PDR7 fractions were tested in the mouse TPA-induced ear oedema model. They decreased oedema by 5.55 ± 2.03 mg, 4.41 ± 2.55 mg, and 6.23 ± 1.7 mg, respectively, representing a reduction of 56.58%, 65.45%, and 51.23%, respectively. They were not significantly different from the effect of treatment with indomethacin (Table 5).

#### Quantification of Pro-Inflammatory the Cytokines IL-10 and TNF-α

In Figure 3A, it is shown that the group of mice that did not receive TPA (Basal) have an IL-10 concentration of 37.5 pg/mg of protein, a value that is significantly higher than the group that received only phorbol ester (* *p* < 0.05). Indomethacin (INDO), as well as treatments derived from *P. laevigata*, significantly increase the values of this pro-inflammatory cytokine when compared with the group without treatment (TPA, * *p* < 0.05), with the treatment with compound **1** being the one that shows the greatest effect on this variable.

Quantification of the pro-inflammatory cytokine TNF-α is shown in Figure 3B. It can be observed that the values of the Basal group, as well as those who received Indo at 1 mg/ear, were statistically lower than the group that received TPA. Treatments with PD, PDR7, and compound **1** were able to significantly decrease not only the oedema but also the concentrations of this pro-inflammatory molecule (* *p* < 0.05).

### 2.5. Antibacterial Activity

Antibacterial activity was determined based on the minimum inhibitory concentration (MIC) of the PH, PD, and PM extracts against 14 bacterial or fungal strains (Table 6). The PM extract presented an antibacterial effect against the 14 evaluated strains. For *S. aureus* ATCC 29213, methicillin-resistant *S. aureus* ATCC 43300, *S. epidermidis* ATCC 35984, *S. epidermidis* ATCC 12228, *S. epidermidis* ATCC 49134, *S. haemolyticus* derived from ATCC 29970, *K. pneumoniae* ATCC 700603, *P. aeruginosa* ATCC 27853, *E. coli* ATCC 25922, *S. dublin* ATCC 9676, *E. cloacae* ATCC 700323, and *C. albicans* ATCC 10231, the MIC was <12.5 µg/mL. For *E. faecalis* ATCC 29212 and *E. coli* ATCC 8739, the MIC values were 50 and 200 µg/mL, respectively. It is worth noting that the PD and PM extracts had MICs of 12.5 and 6.25 µg/mL, respectively, against the methicillin-resistant strain (i.e., methicillin-resistant *S. aureus* ATCC 43300).

### 2.6. Antibacterial Activity of Fractions and Compound **1**

The PMR2, PMR5, PMR6, PMR7, and PMR10 fractions of the PM extract presented antimicrobial activity (Table 7). The PMR6, PMR7, and PMR10 fractions presented an MIC of ≤25 g/mL for 12 of the microorganisms, namely, *S. aureus* ATCC 29213, methicillin-resistant *S. aureus* ATCC 43300, *S. epidermidis* ATCC 35984, *S. epidermidis* ATCC 12228, *S. epidermidis* ATCC 49134*, S. haemolyticus* derived from ATCC 29970, *K. pneumoniae* ATCC 700603, *P. aeruginosa* ATCC 27853, *E. coli* ATCC 25922, *S. dublin* ATCC 9676, *E. cloacae* ATCC 700323, and *C. albicans* ATCC 10231, and an MIC ≥ 25 µg/mL for *E. coli* ATCC 8739.

The PDR7 fraction of the PD extract presented an MIC of ≥25 µg/mL for seven strains, namely, methicillin-resistant *S. aureus* ATCC 43300, *S. epidermidis* ATCC 35984, *E. faecalis* ATCC 29212, *P. aeruginosa* ATCC 27853, *E. coli* ATCC 8739, *S. dublin* ATCC 9676, and *E. cloacae* ATCC 700323, and compound **1** had an MIC ≤ 8 µg/mL for five strains, namely, *K. pneumoniae* ATCC 700603, *P. aeruginosa* ATCC 27853, *E. coli* ATCC 25922, *S. dublin* ATCC 9676, and *C. albicans* ATCC 10231. It is important to mention that the PMR6, PMR7, and PMR10 fractions inhibited the methicillin-resistant strain (methicillin-resistant *S. aureus* ATCC 43300) at concentrations of 6.25 and 25 μg/mL. Gentamicin was used as a positive control at a concentration of 10 µg/mL as it is a first-line drug in the treatment of microbial infections.

## 3. Discussion

*P. laevigata* has been used in Mexico to treat eye diseases, rash, and digestive system disorders. It is also used to treat dysentery, an infectious disease characterised by intestinal inflammation and bloody diarrhoea caused by the *Shigella* bacterium or an amoeba and spread through contaminated food or dirty water [24,25,26]. Based on these data, we selected the aerial parts of this plant and evaluated the biological activity of the PH, PD, and PM extracts.

TPA is a skin irritant, the action of which causes keratinocytes and epidermal dendritic cells to produce inflammatory molecules such as TNF-α, causing acute inflammation with the recruitment of immune cells, namely, neutrophils, macrophages, and mast cells, from the skin tissue [27,28]. The particular mechanism on which TPA exerts its effect is that it induces intracellular signaling pathways through the activation of protein kinase C (PKC), including PI3K/AKT/NF-κB signaling, STAT3 signaling, and the consequent generation of inflammatory mediators such as TNF-α, IL-1β, IL-6, iNOS, COX-2, keratinocyte-derived chemokine, and macrophage inflammatory protein (MIP-2), among other chemokines and prostaglandins which maintain inflammation [29].

The anti-inflammatory activity of *Prosopis juliflora* and *Prosopis alba* extracts has been studied. Researchers have attributed this activity to alkaloids and phenolic compounds present in the aerial parts of these species [30,31]. In the present study, the administration of the PD extract (1 mg/ear) showed the highest percentage of inhibition (75.96%). This activity could be related to the phytol content (70.11%) in the PD extract. Phytol is a diterpene belonging to the group of unsaturated branched-chain alcohols. It has been shown to have several pharmacological properties, such as antinociceptive, anti-inflammatory, and antioxidant properties. This compound reduces plantar inflammation induced by different inflammatory agents. Its anti-inflammatory effect is due to its ability to decrease myeloperoxidase (MPO) activity, the levels of the pro-inflammatory cytokines TNF-α and IL-1β; the concentration of malondialdehyde (MDA) [32], and the release of serotonin, bradykinin, prostaglandin, and histamine [33]. On the other hand, Carvalho et al. [34] showed that phytol inhibits the release of the inflammatory mediators TNF-α, IL-6, and cyclooxygenase 2 (COX2) in complete Freund’s adjuvant–induced rheumatoid arthritis by inhibiting the activation of the p65 pathway of NF-κB through dephosphorylation/inhibition of p38 [34]. It is observed in the present work that PD, PDR7, and compound **1** were able to decrease the levels of TNF-α and increase those of IL-10 in the homogenate of mouse ears exposed to TPA. It can be observed that compound **1** is acting as a regulator of the immune response by causing a greater effect than PD and PDF7 on the concentration of IL-10, a cytokine with potent anti-inflammatory actions that is crucial in the prevention of inflammatory damage processes [35]. While TNF-α is released by dendritic cells and macrophages, its expression is controlled by the activation of TLR4 associated with the regulation of the transcriptional factor NF-κB [36]. It is therefore likely that the effects of **1**, PD, and PDR7 are acting on this signaling pathway, so it is necessary to perform experiments that allow for the verification of this hypothesis. It was also observed that PDF7 has a similar behavior to the non-steroidal anti-inflammatory drug, indomethacin, which was used as a positive control. It is known that this drug inhibits the enzyme Cyclooxygenase 2 (COX-2), causing its anti-inflammatory effect; however, it is also capable of inhibiting COX-1 and inducing oxidative stress, which induces undesirable gastrointestinal effects [37]. The mechanism of action by which PDR7 is exerting the activity in question is unknown. It is expected that its therapeutic capacity will not have the same long-term consequences as indomethacin; for this, it is necessary to continue with a long-term and more refined study to define whether or not it has a toxicity profile.

In this study, microorganisms were selected primarily for their impact on the health sector as they generate resistance to commercial drugs. Several groups of microorganisms are resistant to antibiotics and antimicrobial agents, making the treatment of infectious diseases difficult. Hence, there is an extensive research effort to identify alternative ways to treat infections caused by microorganisms, including plant-based molecules [38]. In the present study, the PM extract inhibited the growth of all strains with good efficacy (MIC 6.25 and 12.5 μg/mL). The PM extract showed stronger antimicrobial activity than the PH and PD extracts. The antibacterial activity of the PM extract is comparable to the results reported by Nava-Solis et al. [19]; their methanolic extract of *P. laevigata* showed strong antimicrobial activity, especially against *S. aureus* (MIC = 0.62 mg/mL) and *E. coli* (MIC = 0.62 mg/mL). Sanchez [39], Khan et al. [40], and Raghavendra et al. [41] presented MICs of <100 µg/mL for different extracts from other *Prosopis* species. Among the 14 tested strains, the Gram-positive bacteria were more sensitive to the extracts. The efficacy of the extracts against Gram-positive bacteria may be attributed to their lack of an outer membrane, rendering them more susceptible to damage caused by active compounds, mainly polyphenols. The anti-fungal activity against Ca is also comparable with results obtained by Nava-Solis et al. [19]. The metabolic content of *P. laevigata* includes phytol, which can increase the accumulation of reactive oxygen species, causing an intracellular imbalance of the antioxidant system of *P. aeruginosa* and thus inhibiting this bacterium [42].

As demonstrated in the bioassays, the extract, fractions, and compounds isolated from *P. laevigata* showed significant effects associated with the inhibition of proinflammatory cytokines and against the growth of 13 pathogenic strains, suggesting potential therapeutic applications for infectious and inflammatory diseases.

This study opens the door to future lines of research and application, particularly in the development of new therapeutic agents. Furthermore, this study provides compelling evidence that *P. laevigata* can be classified as a phytopharmaceutical with pharmacological effects, highlighting its potential to combat microbial diseases and inflammatory processes. Future research should also focus on standardising the extraction process and analysing potential variations in chemical composition due to environmental factors. It is extremely important to validate the therapeutic efficacy and safety of the extract, fractions, EV, and other bioactive compounds to provide a more precise understanding of their toxicity, pharmacokinetics, synergistic effects, and bioavailability. Finally, to determine the clinical relevance of EV as a future candidate for the development of new phytopharmaceuticals, it is crucial to implement human clinical trials.

## 4. Materials and Methods

### 4.1. Plant Material

The aerial parts of *P. laevigata* were collected from the Ricardo Soto 1, las Calaveras, Higuerón (18°35′16″ N, 99°10′35″ W, 895 m above sea level) in Morelos State, Mexico, in December 2019. The sample specimen of this material was identified by M.C. Gabriel Flores at the Autonomous University of Morelos State (voucher code number: 39811); it was stored in the UAEM Herbarium. The fresh plant material of *P. laevigata* was dried at 30 °C.

### 4.2. Preparation of Extracts

Dried material (1.47 kg) was milled in a grinder (particle size 4 mm, Pulvex). The extraction process involved serial maceration with solvents of ascending polarity (*n*-hexane, dichloromethane and methanol) for 24 h. Three extracts (PH, PD, and PM) were evaluated in the anti-inflammatory and antibacterial assays described below. The macerations resulted in the following yields for each extract: 0.33% for PH, 1.49% for PD, and 8.79% for PM.

### 4.3. HPLC–Photo Diode Array (PDA) Analysis of the PM Extract

HPLC-PDA analysis used a Supelcosil LC-F column (4.6 mm × 250 mm internal diameter, 5 µm particle size; Sigma-Aldrich, Bellefonte, PA, USA). The mobile phase consisted of 0.5% trifluoroacetic acid aqueous solution (solvent A) and acetonitrile (solvent B). The gradient system was as follows: 0–1 min, 0% B; 2–3 min, 5% B; 4–20 min, 30% B; 21–23 min, 50% B; 24–25 min, 80% B; 26–27 min, 100% B; 28–30 min, 0% B. The flow rate was maintained at 0.9 mL/min, with a sample injection volume of 10 µL and a wavelength range detection of 190–600 nm. Terpenes were analysed at 220 nm and phenolic compounds at 350 nm.

### 4.4. Identification by GS-MS

The chemical composition of the PH and PD extracts was analysed on an Agilent gas chromatograph equipped with a quadruple mass detector in electron impact mode at 70 eV. Volatile compounds were separated on a HP 5MS (25 m long, 0.2 mm internal diameter, and 0.3 µm film thickness). The oven temperature was set at 40 °C for 1 min, followed by an increase from 40 to 250 °C at 10 °C/min, with maintenance at 250 °C for 20 min. The mass detector conditions were as follows: an interphase temperature of 200 °C and a mass acquisition range of 20–550 *m*/*z*. The injector and detector temperatures were set at 250 and 280 °C, respectively. Spitless injection mode was carried out with 1 µL of each fraction (3 mg/mL solution). The carrier gas was helium at a flow rate of 1 mL/min. The mass spectra were obtained in the time span of 30–35 min. The comparative percentages of each metabolite were estimated by equating their average peak area to the total areas, while the parameters necessary to determine compound classification, such as the molecular weight, formula, and structure of the active metabolites of the test sample, alongside its name, were corroborated. The compounds were identified by comparing the obtained mass spectra with the mass spectra from the literature. This endeavour allowed for the identification of 11 compounds.

### 4.5. Chromatographic Fractionation of the PD Extract

The PD extract (21.1 g) was adsorbed in silica gel and applied to a silica gel gravity column (51 g, 70–230 mesh, Merck, Darmstadt, Germany). A gradient of *n*-hexane/ethyl acetate was used to elute the column with an increase in polarity of 10% per 500 mL. This chromatographic process resulted in 35 fractions, which were grouped into 8 fractions according to the similarity of the constituents (PDR1–PDR8). The anti-inflammatory activity of the PDR3, PDR6, and PDR7 factions were evaluated.

### 4.6. Chromatographic Separation of the PDR7 Fraction to Obtain VE (**1**)

The PDR7 fraction (1.3 g) obtained from the PD extract was purified by successive open column reversed-phase chromatography using RP-18 silica gel (10 g, Sigma-Aldrich, St. Louis, MO, USA) as the stationary phase and a gradient of water/acetonitrile as the mobile phase, yielding 54 fractions with volumes of 10 mL. Thin-layer chromatography showed that subfractions 21 to 27 showed a single spot in the UV spectra (one compound), so they were pooled and analysed by one- and two-dimensional NMR. The compound was identified as VE (see Figure 3). This compound was evaluated as anti-inflammatory and antimicrobial.

### 4.7. Chromatographic Fractionation of the PM Extract

The PM extract (38.8 g) was adsorbed in silica gel and applied to a silica gel gravity column (100 g, 70–230 mesh, Merck, Darmstadt, Germany). A gradient of *n*-hexane/ethyl acetate was used to elute the column with an increase in polarity of 10% per 250 mL. This chromatographic process resulted in 36 fractions, which were grouped into 10 fractions according to the similarity of the constituents (PMR1–PMR10). Based on the chemical content, the antimicrobial activity of the PMR2, PMR5, PMR6, PMR7, and PMR10 fractions was evaluated.

### 4.8. Determination of the Anti-Inflammatory Activity

#### 4.8.1. Experimental Animals

The methodology followed a published protocol [43]. Mice were acquired from Envigo México RMS (Mexico City, Mexico). The mice were used in accordance with NOM-062-ZOO-1999. This protocol was registered with the Ethics and Research Committee (R-2021-1702-010).

#### 4.8.2. Mouse TPA-Induced Ear Oedema Model

Ear oedema was induced following a previously described method [43,44]. The extracts, fractions, and compound **1** were evaluated at a dose of 1 mg/ear. All treatments were dissolved in acetone. Each treatment was applied to the left ear immediately after the administration of TPA; the right was used as a control. The reduction in oedema (in milligrams) and the percentage of inhibition (%) were determined.

#### 4.8.3. Quantification of the Pro-Inflammatory the Cytokines IL-10 and TNF-α

The mouse ears were dissected five hours after applying the treatments and TPA and frozen, further disintegrated into phosphate buffer (pH 7.0) with protease inhibitor (PMFS), and centrifuged for 5 min at 14,000 rpm. The supernatant was collected and stored at @70 °C for the ELISA analysis. In these samples, the cytokines IL-10 and TNF-α were quantified. The measurement technique was performed using a kit OptEIATM ELISA sets; BD Biosciences, Franklin Lakes, NJ, USA) according to the manufacturer’s instructions. Briefly, we added 100 μL/well of the antibody uptake to 96-well plates; and the plates were incubated for 12 h at 4 °C. Then, the dish was washed with PBS (phosphate- buffered saline) solution (0.05% of Tween-20, 300 μL/well × three times). A total of 100 μL of PBS with fetal bovine serum (FBS) was added at 10%, pH 7.0, for 1 h at room temperature. The contents were discarded, and the plate was washed with PBS buffer (0.05% of Tween-20, 300 μL/well × three times). To the corresponding wells we added 100 μL of the standard, the target (PBS with PBS), and the test samples. The plate was incubated for 2 h at room temperature. The contents were discarded, and the plate was washed with PBS buffer (0.05% of Tween-20, 300 μL/well × five times). A total of 100 μL/well of a detection antibody and a streptavidin–horseradish peroxidase (HRP) enzyme solution was added. These plates were incubated for 1 h and washed with 300 μL/well × seven times with a PBS solution (combined with 0.05% of Tween-20).

### 4.9. Antimicrobial Activity

#### 4.9.1. Microorganisms

Antibacterial activity was determined by the microdilution method using 7 Gram-positive strains (*Staphylococcus aureus* ATCC 29213, methicillin-resistant *Staphylococcus aureus* ATCC 43300, *Staphylococcus epidermidis* ATCC 35984, *Staphylococcus epidermidis* ATCC 12228, *Staphylococcus epidermidis* ATCC 49134, methicillin-resistant *Staphylococcus haemolyticus* derived from ATCC 29,970, and *Enterococcus faecalis* ATCC 29212), 6 Gram-negative strains (*Klebsiella pneumoniae* ATCC 700603, *Pseudomonas aeruginosa* ATCC 27853, *Escherichia coli* ATCC 8739, *Escherichia coli* ATCC 25922, *Salmonella dublin* ATCC 9676, and *Enterobacter cloacae* ATCC 700323), and one yeast (*Candida albicans* ATCC 10231). The microorganisms were provided by the Universidad Autónoma de Guerrero (UAGro). Two microliters of each strain were inoculated and incubated under constant temperature (20 ± 2 °C) and humidity (50 ± 5%) conditions.

The strains were maintained on Trypticase soy agar (Merck) at 37 °C for 24 h.

#### 4.9.2. MIC

The MIC was determined based on a published method [45], with some changes in concentrations. All bacteria were standardised to 0.5 McFarland units (1.5 × 10^8^ colony-forming units [CFU]/mL). Extracts and fractions were diluted with 200 μL of DMSO and 800 μL of H_2_O. Concentrations of 25, 50, 100, and 200 μg/mL were tested. Two microlitres of each strain were inoculated and incubated under controlled temperature (37 ± 2 °C) and humidity (13 ± 2%) conditions at 37 °C for 24 h. Gentamicin was used as a positive control (10 μg/mL), and DMSO (2%) was used as a negative control. All experiments were performed in triplicate.

### 4.10. Statistical Analysis

The data are expressed as the mean ± standard error of the mean (SEM). Statistical significance was determined using analysis of variance (ANOVA) followed by Dunnett’s and Bonferroni tests to compare each treatment with the vehicle and indomethacin groups. A *p*-value ≤ 0.05 was considered to indicate a statistically significant difference.

## 5. Conclusions

The PD extract of *Prosopis laevigata* showed the strongest anti-inflammatory effect, and the PM extract exerted the most pronounced antimicrobial effect. In the PD extract, compound **1** (VE) was isolated and identified and could be one of the compounds responsible for the anti-inflammatory effect according to the mouse TPA-induced ear oedema model and the observed concentration of pro-inflammatory cytokines IL-10 and TNF-α. On the other hand, flavonoid-type compounds present in the PM extract and its fractions may be responsible for the antibacterial activity.

## Figures and Tables

**Figure 1 plants-14-01118-f001:**
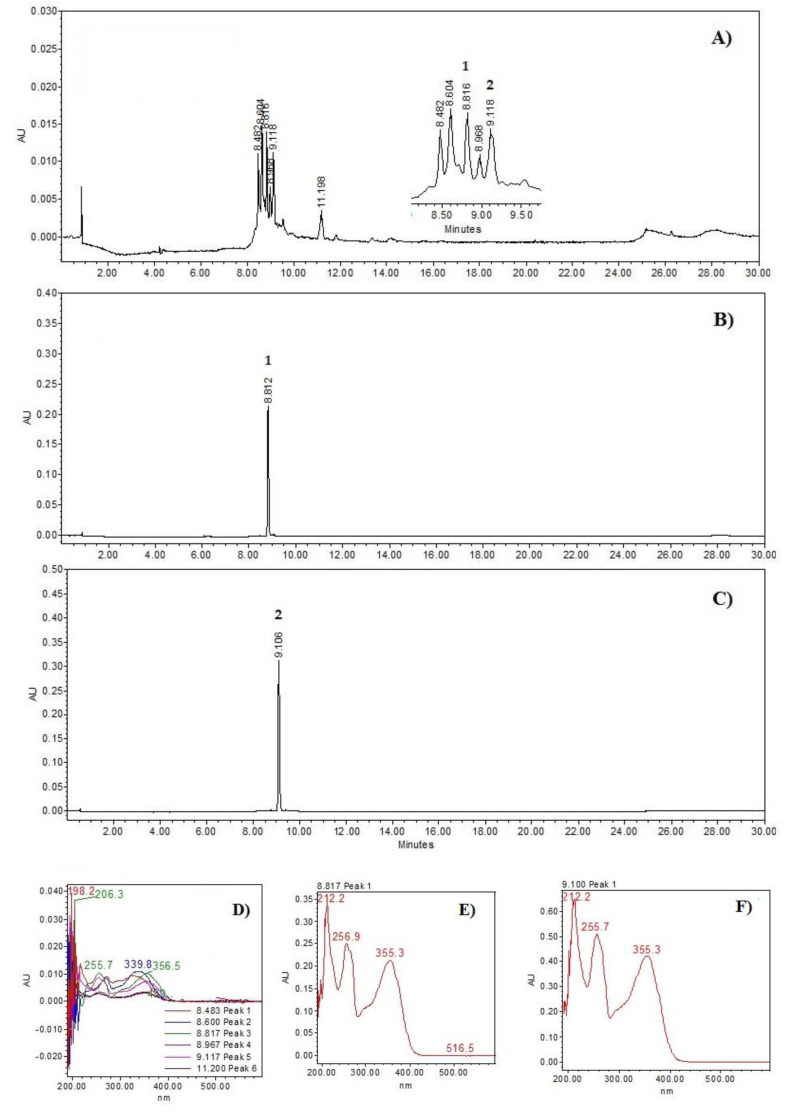
Chemical profiles: (**A**) *Prosopis laevigata* methanolic extract at 350 nm; (**B**) rutin (**2**); (**C**) quercetin 3-*O*-glucoside (**3**). UV spectrum: (**D**) *Prosopis laevigata* methanolic extract; (**E**) rutin; (**F**) quercetin 3-*O*-glucoside (**3**).

**Figure 3 plants-14-01118-f003:**
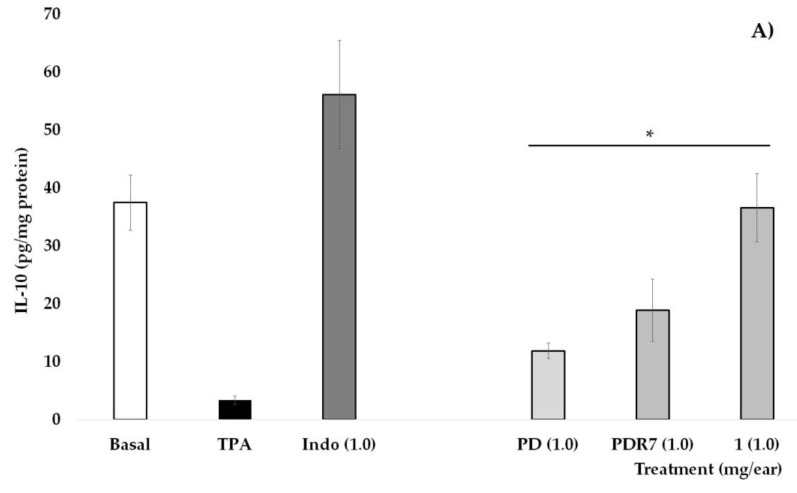
Effects of different *P. laevigata* treatments—i.e., dichloromethane extract (PD), fractions from this extract (PDR7), and compound **1**—on the concentration of IL-10 (**A**) and TNF-α (**B**) on the ears of mice stimulated with a phorbol ester (TPA). Graphical representation showing media ± SD (n = 5) with one-way ANOVA and post hoc Bonferroni (* *p* ≤ 0.05) when compared with the TPA group.

**Table 1 plants-14-01118-t001:** The gas chromatography–mass spectrometry profile of the *Prosopis laevigata n*-hexane extract.

Peak No	Name	Retention Time (min)	Relative Area (%)	Mw (g/mol)	Type of Compound
1	1,54-Dibromotetrapentacontane	21.37	4.27	917.2	Alkane
2	Hexanedioic acid	22.22	5.64	370.6	Fatty acid
3	Tetracontane	23.13	3.47	563.1	Alkane
4	6,6′-di-tert-butyl-4,4′-diethyl-2,2′-methylenediphenol	23.36	9.64	368.55	Alkylated phenol
5	Heptacosane	24.24	2.72	380.7	Alkane
6	Tetratetracontane	25.64	3.68	619.2	Alkane
7	Squalene	27.89	1.48	410.73	Triterpene

**Table 2 plants-14-01118-t002:** The gas chromatography–mass spectrometry profile of the *Prosopis laevigata* dichloromethane extract.

Peak No.	Name	Retention Time (min)	Relative Area (%)	Mw (g/mol)	Type of Compound
1	6-Hydroxy-4,4,7a-trimethyl-5,6,7,7a-tetrahydrobenzofuran-2(4H)-one	13.77	12.87	196.24	Benzofuran
2	Phytol	19.77	70.11	296.53	Diterpene alcohol
3	Bis[(2S)-2-ethylhexyl] hexanedioate	22.22	17.0	370.6	Fatty acid

**Table 3 plants-14-01118-t003:** Anti-inflammatory activity of the *Prosopis laevigata* extracts.

Treatment	Oedema (mg)	Oedema Inhibition (%)
VEH	11.43 ± 0.87	–
INDO	4.00 ± 1.40 *	65.01
PH	4.48 ± 1.83 *	60.81
**PD**	**2.74 ± 0.68** *****	**75.96**
PM	4.54 ± 1.41 *	60.29

Abbreviations: INDO, indomethacin; PD, *P. laevigata* dichloromethane extract; PH, *P. laevigata n*-hexane extract; PM, *P. laevigata* methanolic extract; VEH, vehicle. * *p* ≤ 0.05, Anova post hoc Dunnet in comparison with vehicule. Highlighted in bold indicates greater effect.

**Table 5 plants-14-01118-t005:** Anti-inflammatory activity of the fractions and compound (**1**) from the dichloromethane extract of *Prosopis laevigata*.

Treatment	Oedema (mg)	Oedema Inhibition (%)
VEH	12.78 ± 2.64	–
INDO	4.00 ± 1.40 *	65.01
PDR3	5.55 ± 2.03 *	56.58
PDR6	4.41 ± 2.55 *	65.45
**PDR7**	6.23 ± 1.70 *	**51.23**
**VE (1)**	1.90 ± 0.63 *	**85.13**

Abbreviations: **1**, ethyl veratrate; INDO, indomethacin; PDR, fractions of the *P. laevigata* dichloromethane extract; VEH, vehicle. * *p* ≤ 0.05, Anova post hoc Dunnet in comparison with vehicule. Highlighted in bold indicates greater effect.

**Table 6 plants-14-01118-t006:** Antimicrobial activity of the *Prosopis laevigata* extracts.

Bacterial Clinical Isolate	Minimum Inhibitory Concentration (μg/mL)		
	PH	PD	PM
*Staphylococcus aureus* ATCC 29213	200	200	6.25
*Methicillin-resistant Staphylococcus aureus* ATCC 43300	200	12.5	6.25
*Staphylococcus epidermidis* ATCC 35984	200	12.5	6.25
*Staphylococcus epidermidis* ATCC 12228	50	6.25	6.25
*Staphylococcus epidermidis* ATCC 49134	-	200	6.25
*Staphylococcus haemolyticus* derived from ATCC 29970	-	200	6.25
*Enterococcus faecalis* ATCC 29212	6.25	6.25	6.25
*Klebsiella pneumoniae* ATCC 700603	6.25	12.5	6.25
*Pseudomonas aeruginosa* ATCC 27853	6.25	12.5	6.25
*Escherichia coli* ATCC 8739	-	-	12.5
*Escherichia coli* ATCC 25922	6.25	25	12.5
*Salmonella dublin* ATCC 9676	6.25	6.25	6.25
*Enterobacter cloacae* ATCC 700323	12.5	12.5	12.5
*Candida albicans* ATCC 10231	6.25	6.25	6.25

The values correspond to the average (*n* = 3). The following organisms were evaluated: seven Gram-positive strains, six Gram-negative strains, and one yeast. The positive control was gentamicin (10 µg/mL), and the negative control was 2% dimethyl sulfoxide. Abbreviations: PD, *P. laevigata* dichloromethane extract; PH, *P. laevigata n*-hexane extract; PM, *P. laevigata* methanolic extract.

**Table 7 plants-14-01118-t007:** Antibacterial activity of fractions of the methanolic extract (PMR2, PMR5, PMR6, PMR7, PMR10), the dichloromethane extract (PDR7), and compound **1** of *Prosopis laevigata*.

Bacterial Clinical Isolate	Minimum Inhibitory Concentration (μg/mL)						
	Fractions						
	PMR2	PMR5	PMR6	PMR7	PMR10	PDR7	1
*Staphylococcus aureus* ATCC 29213	-	-	6.25	25	6.25	-	-
Methicillin-resistant *Staphylococcus aureus* ATCC 43300	-	-	6.25	25	12.5	100	-
*Staphylococcus epidermidis* ATCC 35984	25	-	6.25	6.25	6.25	25	-
*Staphylococcus epidermidis* ATCC 12228	200	-	6.25	25	25	-	-
*Staphylococcus epidermidis* ATCC 49134	-	-	6.25	12.5	12.5	-	-
*Staphylococcus haemolyticus* derived from ATCC 29970	-	-	6.25	12.5	12.5	-	-
*Enterococcus faecalis* ATCC 29212	25	25	6.25	6.25	50	100	-
*Klebsiella pneumoniae* ATCC 700603	25	-	6.25	6.25	6.25	-	2
*Pseudomonas aeruginosa* ATCC 27853	25	25	6.25	25	6.25	100	8
*Escherichia coli* ATCC 8739	-	-	25	200	50	200	-
*Escherichia coli* ATCC 25922	25	25	12.5	25	25	-	8
*Salmonella dublin* ATCC 9676	6.25	6.25	6.25	6.25	6.25	25	2
*Enterobacter cloacae* ATCC 700323	12.5	12.5	12.5	12.5	12.25	25	-
*Candida albicans* ATCC 10231	6.25	6.25	6.25	6.25	6.25		2

The values correspond to the average (*n* = 3). The following organisms were evaluated: seven Gram-positive strains, six Gram-negative strains, and one yeast. The positive control was gentamicin (10 µg/mL), and the negative control was 2% dimethyl sulfoxide.

## Data Availability

Data is contained within the article or Appendix A.

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
