# Peer review of "Chemical Profile Analysis of Prosopis laevigata Extracts and Their Topical Anti-Inflammatory and Antibacterial Activities"

_plants, 2025, doi:10.3390/plants14071118_

Round 1

Reviewer 1 Report (Previous Reviewer 2)

Comments and Suggestions for Authors

After reviewing the manuscript, I can confirm that the key comments from the reviewers have been addressed, and I agree with the changes made. I believe the text is now in a significantly improved format and, with some minor corrections, can be recommended for publication in MDPI/Plants.

I have only a few minor comments on the text:

  1. L226 – unnecessary parentheses
  2. L227 – inconsistent notation throughout the text – "Methicillin-resistant.." vs. "methicillin-resistant" (see L198, L227, etc.)
  3. Tab7 – incorrect format, the table structure is not displayed
  4. Tab7 – error in the designation of strain ATCC 29970

Author Response

I have only a few minor comments on the text:

  1. L226 – unnecessary parentheses
  2. L227 – inconsistent notation throughout the text – "Methicillin-resistant.." vs. "methicillin-resistant" (see L198, L227, etc.)
  3. Tab7 – incorrect format, the table structure is not displayed
  4. Tab7 – error in the designation of strain ATCC 29970

Response: Dear reviewer, we appreciate your time reviewing the document, which helped improve it. All suggestions were addressed in the document

Reviewer 2 Report (Previous Reviewer 3)

Comments and Suggestions for Authors

There are numerous flaws in the entire study. Some of them are as follows:

1) Why was only one particular dose of infection selected?

2) After how many hours were the samples extracted for cytokine analysis?

3) Why was acetone used as a solvent instead of DMSO?

4) What was the reason for selecting only certain types of cytokines?

5) What dose of infection was administered?

Overall, the discussion, materials and methods, and results sections are very weak, and nothing is written properly. Therefore, I believe the study is not suitable for publication.

Comments on the Quality of English Language

The English language needs improvement throughout the manuscript. There is no flow while reading the manuscript, making it unengaging for the reader.

Author Response

Reviewer

There are numerous flaws in the entire study. Some of them are as follows:

1) Why was only one particular dose of infection selected?

Response: Dear reviewer, perhaps you are referring to inflammation, and this dose was used because anti-inflammatory activity has been observed in this pharmacological model, and many other authors have used it. Furthermore, a dose was used to minimize the number of mice, as suggested by the ethics committee.

https://doi.org/10.1080/10667857.2019.1638671

https://doi.org/10.1002/(SICI)1099-1573(199811)12:7<484::AID-PTR341>3.0.CO;2-L

https://doi.org/10.1016/j.bbrc.2016.09.009

2) After how many hours were the samples extracted for cytokine analysis?

Response: The samples were extracted after the treatments and TPA were applied, which was after five hours. It was added to the methodology.

3) Why was acetone used as a solvent instead of DMSO?

Response: The manufacturer reports that TPA is soluble in acetone, DMSO, ethyl acetate, ethanol and methylene chloride. (https://www.sigmaaldrich.com/deepweb/assets/sigmaaldrich/product/documents/816/596/p8139pis.pdf). In our experience, based on other publications in the field, we performed the procedure with acetone, and the results indicate that the level of edema is similar in all cases. It should be noted that there are reports indicating that dimethyl sulphoxide (DMSO), for example, can cause acute irritation of the mouse ear. (Laihia, J. K., Taimen, P., Kujari, H., & Leino, L. (2012). Topical cis-urocanic acid attenuates oedema and erythema in acute and subacute skin inflammation in the mouse. British Journal of Dermatology, 167(3), 506-513.)

4) What was the reason for selecting only certain types of cytokines?

Response: In a 2006 study, the involvement of TNF-α in TPA-induced skin edema was measured, and it was shown that administration of this phorbol ester caused gradual edema over 24 hours with a maximum peak at that point. When TNF-α levels were quantified at different times, a maximum peak of this cytokine was observed at 5 hours. The authors were agree that the participation of this pro-inflammatory molecule during TPA-induced edema is relevant and that they agree with data from other authors indicating that TNF-α-deficient mice do not develop skin inflammation after TPA administration (Murakawa, M., Yamaoka, K., Tanaka, Y., & Fukuda, Y. (2006). Involvement of tumor necrosis factor (TNF)-α in phorbol ester 12-O-tetradecanoylphorbol-13-acetate (TPA)-induced skin edema in mice.Biochemical pharmacology, 71(9), 1331-1336.) (Moore RJ, Owens DM, Stamp G, Arnott C, Burke F, East N, et al. Mice deficient in tumor necrosis factor-alpha are resistant to skin carcinogenesis. Nat Med 1999;5:828–31).

Also, the skin receiving TPA presents a significant irritation, which is enhanced in animals deficient in IL-10 or in mice deficient in myeloid-specific IL-10, but not in mice deficient in IL-10 derived from T cells. Thus, the innate Th1 type response is subject to regulation by IL-10. Both arguments, presented above, led us to propose IL-10 and TNF-α as part of the evaluation of the effects of the medicinal species on local inflammation by TPA (Berg, D. J., Leach, M. W., Kühn, R., Rajewsky, K., Müller, W., Davidson, N. J. and Rennick, D., Interleukin 10 but not interleukin 4 is a natural suppressant of cutaneous inflammatory responses. J. Exp. Med. 1995. 182: 99–108).

5) What dose of infection was administered?

Response: For both inflammation and antimicrobial agents, administration doses were used in accordance with those reported in the literature.

In general, the discussion, materials and methods, and results sections are very deficient, and nothing is written correctly. Therefore, I believe the study is not suitable for publication.

Response: Based on your comment, we believe this work was conducted using established techniques for the field of natural products that have already been validated. Therefore, we believe this work represents an important contribution to basic science because the objective is to conduct preclinical studies with standardized extracts based on the active ingredient, such as this study on ethyl veratrate, in patients with these conditions (inflammation and microbial), which are common and important in the healthcare field.

Reviewer 3 Report (New Reviewer)

Comments and Suggestions for Authors

Dear Authors,

Your study “Chemical Profile Analysis of Prosopis laevigata Extracts and Their Topical Anti-Inflammatory and Antibacterial Activities” presents valuable insights into the anti-inflammatory and antimicrobial potential of Prosopis laevigata. The combination of phytochemical analysis (HPLC, GC-MS, NMR) and biological assays (cytokine quantification, MIC determination, and inflammation models) provides a comprehensive approach to validating the plant’s medicinal properties.

While the manuscript is well-structured and scientifically sound, some areas require improvement for clarity, reproducibility, and a stronger discussion of implications. Below are some major and minor suggestions for revisions:

The selection of extract concentrations (1 mg/ear in the TPA-induced inflammation model) should be justified—was this based on prior studies, preliminary experiments, or a dose-response curve?

The rationale for selecting specific bacterial strains should be explained. Were these chosen based on clinical relevance, prior research on Prosopis species, or susceptibility to plant-derived antimicrobials?

Provide details on incubation conditions (temperature, humidity) for bacterial assays to ensure reproducibility.

Some figures (e.g., HPLC chromatograms, cytokine quantification graphs) could be enhanced in resolution and contrast for better readability.

Tables presenting MIC values and cytokine levels could be highlighted with bold formatting or color differentiation to make key findings more visually accessible.

Ensure that all chemical structure images are clear and properly labeled.

The MIC values for antibacterial activity should be compared to standard antibiotics (e.g., gentamicin) to provide a clearer understanding of their clinical relevance.

The discussion on anti-inflammatory activity could benefit from a comparison with known NSAIDs (e.g., indomethacin) to assess the potential strength of ethyl veratrate.

A section on practical applications would enhance the impact of the study—how could P. laevigata be formulated into medicinal, cosmetic, or nutraceutical products?

The manuscript is generally well-written but contains some complex and lengthy sentences. Consider restructuring certain parts for better readability.

  • Example revision:
    • "This physiological process represents a local natural defence by the immune system in response to damage caused by harmful agents to the body’s cells and tissues."
    • "Inflammation is a natural immune response to tissue damage caused by harmful agents."A thorough proofreading or professional language editing is recommended.

Explain how the purity of isolated ethyl veratrate (VE) was verified—was it tested for impurities using additional spectroscopic methods?

Were residual solvents completely removed from the extracts before biological testing? Any trace amounts could influence results.

The manuscript would benefit from a short paragraph on study limitations (e.g., in vivo confirmation, long-term stability of extracts).

A brief discussion on future directions (e.g., mechanistic studies on ethyl veratrate, toxicity testing, large-scale extraction methods) would strengthen the impact of the findings.

Comments on the Quality of English Language

Minor revision

Author Response

Dear Authors,

Your study “Chemical Profile Analysis of Prosopis laevigata Extracts and Their Topical Anti-Inflammatory and Antibacterial Activities” presents valuable insights into the anti-inflammatory and antimicrobial potential of Prosopis laevigata. The combination of phytochemical analysis (HPLC, GC-MS, NMR) and biological assays (cytokine quantification, MIC determination, and inflammation models) provides a comprehensive approach to validating the plant’s medicinal properties.

While the manuscript is well-structured and scientifically sound, some areas require improvement for clarity, reproducibility, and a stronger discussion of implications. Below are some major and minor suggestions for revisions:

The selection of extract concentrations (1 mg/ear in the TPA-induced inflammation model) should be justified—was this based on prior studies, preliminary experiments, or a dose-response curve?

Response: The selection of the 1 mg/ear dose was based on data on the effect of different pharmacological agents such as indomethacin, ibuprofen, aspirin, piroxicam, among others, which were evaluated in the TPA trial and which has also been a strategy applied in works that the research group has published (Gábor M. 12-O-tetradecanoylphorbol-13-acetate ear oedema (TPA). In: Mouse Ear Inflammation Models and their pharmacological Applications. Akakémiai Kiadó H-1519. Budapest, P.O. 2000, Hungary. Pp 33).

The rationale for selecting specific bacterial strains should be explained. Were these chosen based on clinical relevance, prior research on Prosopis species, or susceptibility to plant-derived antimicrobials?

Response: A paragraph was added to the discussion of the document.

Provide details on incubation conditions (temperature, humidity) for bacterial assays to ensure reproducibility.

Response: The requested data were added to the antimicrobial pharmacology methodology. Humidity was not measured, but it is considered to be 13–15%, depending on the site.

Some figures (e.g., HPLC chromatograms, cytokine quantification graphs) could be enhanced in resolution and contrast for better readability.

Response: The graphics were improved for better visibility in the document.

Tables presenting MIC values and cytokine levels could be highlighted with bold formatting or color differentiation to make key findings more visually accessible.

Response: We appreciate your suggestion; important data have been highlighted in bold throughout the document.

Ensure that all chemical structure images are clear and properly labeled.

Response: The chemical structures have been corrected.

The MIC values for antibacterial activity should be compared to standard antibiotics (e.g., gentamicin) to provide a clearer understanding of their clinical relevance.

Response: The data were compared with gentamicin.

The discussion on anti-inflammatory activity could benefit from a comparison with known NSAIDs (e.g., indomethacin) to assess the potential strength of ethyl veratrate.

Response: A paragraph was added to the discussion section of the document highlighting the potency of ethyl veratrate.

A section on practical applications would enhance the impact of the study—how could P. laevigata be formulated into medicinal, cosmetic, or nutraceutical products?

Response: It's a plant that still requires further study to further emphasize its potential and adequately support its use. However, based on its anti-inflammatory properties, it can be considered a "Phytomedicine based on a standardized extract of the isolated compound and formulated as an ointment or gel with therapeutic potential for skin inflammation."

The manuscript is generally well-written but contains some complex and lengthy sentences. Consider restructuring certain parts for better readability.

  • Example revision:
    • "This physiological process represents a local natural defence by the immune system in response to damage caused by harmful agents to the body’s cells and tissues."
    • "Inflammation is a natural immune response to tissue damage caused by harmful agents." A thorough proofreading or professional language editing is recommended.

Response: Thank you very much for your recommendation. We have therefore reviewed the document and improved the wording of the paragraphs for better readability. The document that was sent for proofreading is attached. It will be resubmitted.

Explain how the purity of isolated ethyl veratrate (VE) was verified—was it tested for impurities using additional spectroscopic methods?

Response:  One- and two-dimensional nuclear magnetic resonance (NMR) (supplementary data) allows us to identify the purity of the compound. If it is mixed, more signals should appear that do not correspond to the identified compound. Additionally, HPLC and mass spectroscopy also provide information on its purity.

Were residual solvents completely removed from the extracts before biological testing? Any trace amounts could influence results.

Response: Solvents were completely removed by using a rotary evaporator and subsequently by a vacuum freeze-drying machine, since solvents can influence the activity of the extract, fraction or compound.

The manuscript would benefit from a short paragraph on study limitations (e.g., in vivo confirmation, long-term stability of extracts).A brief discussion on future directions (e.g., mechanistic studies on ethyl veratrate, toxicity testing, large-scale extraction methods) would strengthen the impact of the findings.

Response: A brief discussion was added to the paper. We appreciate your thorough review, which improved the submitted paper

Round 2

Reviewer 2 Report (Previous Reviewer 3)

Comments and Suggestions for Authors

After reading the justification provided by the authors, I am still not convinced by any of their arguments. In response to almost every question I have raised, instead of providing a proper justification, they have merely cited others' work without offering a clear rationale for conducting the experiment in the manner they did. This suggests to me that they did not conduct a thorough literature review before planning this study. Based on these circumstances, I am not in favor of publishing this article in the journal.

Comments on the Quality of English Language

Still remains the same nothing much has been changed.

This manuscript is a resubmission of an earlier submission. The following is a list of the peer review reports and author responses from that submission.

Round 1

Reviewer 1 Report

Comments and Suggestions for Authors

This paper stated the serious issue antimicrobial resistance by a medicinal plant entitled "Anti-inflammatory and antibacterial activities of Prosopis laevigata".

The study is on hot topic of inflammation and antimicrobial resistance. 

In this paper author pefromed chemical scrrening and followed by in vivo studies. 

My comments are as follows;

1. The introduction requires serious revision, as the background of antimicrobial resistance and its mechanism should be stated. Additionally, the role of medicinal plants in preventing antimicrobial resistance and inflammation and plants' traditional perspective is lacking. 

2. What is TPA please provide full name and then abbreviation. 

3. In in vivo inflammatory studies, I suggest authors provide pictures of edema showing recovery following treatment with plant extract. 

4. What about the serum inflammatory cytokine levels?

5. In antibacterial activity, I would suggest performing a zone of inhibitory study and providing pictures of petri plates showing the zone of inhibition of bacteria following treatment with plant extract. 

6. Does the plant extract inhibit the activation of the MAPKs pathway or cytokines including TNFa, IL-6, PGE2 or MIP1-beta?

7. Anti-inflammatory activities in-vitro should also be conducted as indicated in the following papers: 

Rajpoot, Sehrish Rana, et al. "Study of anti-inflammatory and anti-arthritic potential of curcumin-loaded Eudragit L100 and hydroxy propyl methyl cellulose (HPMC) microparticles." Polymer Bulletin 81.5 (2024): 4335-4350.

Farooq, Sundas, et al. "Preliminary phytochemical analysis: In-Vitro comparative evaluation of anti-arthritic and anti-inflammatory potential of some traditionally used medicinal plants." Dose-Response 20.1 (2022): 15593258211069720.

8. Last paragraph of the discussion, Regarding anti-inflammatory activity, it has been shown that oral administration of 193 50 mg/kg of Phytol in Female albino Swiss mice with rheumatoid arthritis induced with 194 complete Freund’s adjuvant (CFA) can decrease joint inflammation and spinal cord IL-6 195 and COX-2 levels, through downregulated the p38MAPK and NFκB signaling pathways 196 [24].

the author should clearly state that their extract has Phytol and Phytol has been reported to downregulate the P38MAPK and NfKB pathway. Therefore, the anti-inflammatory activity of this extract might be due to phytol abundance thereby reducing the edema. 

Relevance to the results and discussion in the context of the molecular mechanism is lacking in this paper. 

Comments on the Quality of English Language

There are some grammatical and spelling mistakes in the paper. English revision is highly recommended. 

Author Response

Reviewer 3

This paper stated the serious issue antimicrobial resistance by a medicinal plant entitled "Anti-inflammatory and antibacterial activities of Prosopis laevigata".

The study is on hot topic of inflammation and antimicrobial resistance. 

In this paper author pefromed chemical scrrening and followed by in vivo studies. 

My comments are as follows;

1.-The introduction requires serious revision, as the background of antimicrobial resistance and its mechanism should be stated. Additionally, the role of medicinal plants in preventing antimicrobial resistance and inflammation and plants' traditional perspective is lacking. 

Response: Each of the suggestions and recommendations were added to the document in the introduction.

2.- What is TPA please provide full name and then abbreviation.

Response: The abbreviation TPA was defined.

3. In in vivo inflammatory studies, I suggest authors provide pictures of edema showing recovery following treatment with plant extract

Response: Your recommendation is understandable but when carrying out the experiments no photographs were taken pictures of edema showing recovery following treatment with plant extract

4.- What about the serum inflammatory cytokine levels?

Response: In this work on the study of Prosopis laevigata, the objective was to perform a chemical characterization based on two assays, the antimicrobial and the topical local edema assay caused by TPA because it is recognized that this substance causes all the processes associated with inflammation and that the thickness of the ear is the first variable analyzed as a measure of said event, and because this is the first pharmacological approach to this Mexican species and we observed a significant inhibition of edema, an inflammation study is being planned to evaluate its chronic effect and thus be able to measure associated biochemical parameters.

5.- . In antibacterial activity, I would suggest performing a zone of inhibitory study and providing pictures of petri plates showing the zone of inhibition of bacteria following treatment with plant extract. 

Response: Thank you very much for the suggestion. It is considered for the following works because the technique must be established and the inhibition halos can be measured.

6.- Does the plant extract inhibit the activation of the MAPKs pathway or cytokines including TNFa, IL-6, PGE2 or MIP1-beta?

Response: So far, in this work we have only made a first approach as an anti-inflammatory with the TPA model. But we are considering carrying out a more specific method that allows us to see if there is activation of cytokines.

7.- Anti-inflammatory activities in-vitro should also be conducted as indicated in the following papers

Rajpoot, Sehrish Rana, et al. "Study of anti-inflammatory and anti-arthritic potential of curcumin-loaded Eudragit L100 and hydroxy propyl methyl cellulose (HPMC) microparticles." Polymer Bulletin 81.5 (2024): 4335-4350.

Farooq, Sundas, et al. "Preliminary phytochemical analysis: In-Vitro comparative evaluation of anti-arthritic and anti-inflammatory potential of some traditionally used medicinal plants." Dose-Response 20.1 (2022): 15593258211069720.

Response: Thank you for your recommendation. We will consider this technique for future research.

8.- Last paragraph of the discussion, Regarding anti-inflammatory activity, it has been shown that oral administration of 50 mg/kg of Phytol in Female albino Swiss mice with rheumatoid arthritis induced with complete Freund’s adjuvant (CFA) can decrease joint inflammation and spinal cord IL-6 195 and COX-2 levels, through downregulated the p38MAPK and NFκB signaling pathways 196 [24].

Response: Data obtained from the literature.

9. The author should clearly state that their extract has Phytol and Phytol has been reported to downregulate the P38MAPK and NfKB pathway. Therefore, the anti-inflammatory activity of this extract might be due to phytol abundance thereby reducing the edema. 

Response: A paragraph was added to the discussion highlighting that phytol may be responsible for the observed effect.

10. Relevance to the results and discussion in the context of the molecular mechanism is lacking in this paper. 

Response: Thank you very much for your comment. We are aware that the mechanism must be tested and identified, and methodologies are being proposed to demonstrate this. This study was focused on identifying the possible active metabolites of Prosopis laevigata. That is why the title of the work has been changed.

Reviewer 2 Report

Comments and Suggestions for Authors

Dear Authors,

Submitted manuscript number plants-3253435 entitled "Anti-inflammatory and antibacterial activities of Prosopis laevigata" discusses the interesting topic of pro-inflammatory and overall antibacterial potential of natural substances. I consider the focus and the research goal of the study to be adequate and bring quite interesting information for readers. However, the text contains a significant amount of inconsistencies throughout the text as well as inappropriate or inaccurate statements. Thus, it is not even possible to properly focus on the expertise of the study as such. I therefore recommend going through and revising the entire text.

I'll randomly mention a few specific comments below:

1. L116-118 - rewrite the sentence "All the extracts were active....". How active? It is neither appropriate nor professionally correct to specify in this way.

2. L130 italics of Latin names

3. I believe that the designation of sub-chapter 4.5 is not the most appropriate (reference (1) to a specific compound. Please try to set another more suitable title for the chapter to the overall nature of the study. Furthermore, it is not entirely clear from this passage which compound it is, or what peak it is on Fig1, etc.??

4. Chapter 4.9 must be completely revised and written properly and correctly!

- ...were evaluated to inhibit....

- Methicillin-Resistant Staphylococcus aureus (opposite order preferred)

- mislabeling staphylococci on L291!

- L292 and 294 - what does "y" mean in the text??

- L296 redundant dot

- L296 the last part of the text is not a sentence!!, inappropriate notation without any formulation.

- Are these collection trunks? Then, there is a lack of proper identification of the strains. These are some isolates from a previous study, so there is no information about the origin, etc.

- L299 missing superscript "8". However, the whole statement is inappropriately written and, in principle, does not make sense from a professional point of view..).

5. Table 6 - the legend must be modified per the comment above.

6. etc.

Comments on the Quality of English Language

I recommend having a native English speaker review the text.

Author Response

Submitted manuscript number plants-3253435 entitled "Anti-inflammatory and antibacterial activities of Prosopis laevigata" discusses the interesting topic of pro-inflammatory and overall antibacterial potential of natural substances. I consider the focus and the research goal of the study to be adequate and bring quite interesting information for readers. However, the text contains a significant amount of inconsistencies throughout the text as well as inappropriate or inaccurate statements. Thus, it is not even possible to properly focus on the expertise of the study as such. I therefore recommend going through and revising the entire text.

Response: We appreciate your comments. The document was carefully reviewed and corrected according to your suggestion.

1.- L116-118 - rewrite the sentence "All the extracts were active....". How active? It is neither appropriate nor professionally correct to specify in this way.

Response: The text in the document was modified. Results Section 2.5

2.- L130 italics of Latin names

Response: Corrected in the document

3.- I believe that the designation of sub-chapter 4.5 is not the most appropriate (reference (1) to a specific compound. Please try to set another more suitable title for the chapter to the overall nature of the study. Furthermore, it is not entirely clear from this passage which compound it is, or what peak it is on Fig1, etc.??

Response: Change was made to subheading 4.5 and the wording of the paragraph was modified to better describe compound 1, which is ethyl veratrate (VE).

4.- Chapter 4.9 must be completely revised and written properly and correctly!

- ...were evaluated to inhibit....

- Methicillin-Resistant Staphylococcus aureus

(opposite order preferred)

- mislabeling staphylococci on L291!

Methicillin-Resistant Staphylococcus aureus (opposite order preferred)

- mislabeling staphylococci on L291!

- L292 and 294 - what does "y" mean in the text??

- L296 redundant dot

- L296 the last part of the text is not a sentence!!, inappropriate notation without any formulation.

- Are these collection trunks? Then, there is a lack of proper identification of the strains. These are some isolates from a previous study, so there is no information about the origin, etc.

- L299 missing superscript "8". However, the whole statement is inappropriately written and, in principle, does not make sense from a professional point of view..).

Response: Each of the suggestions and recommendations were corrected in the document

5.- Table 6 - the legend must be modified per the comment above.

Response: The table was reviewed and corrected according to the suggestion

Reviewer 3 Report

Comments and Suggestions for Authors

The manuscript has many flaws that cannot be corrected without repeating the experiments. Some of them are as follows:

1)The manuscript claims an anti-inflammatory effect starting from the title, but the authors base this conclusion solely on weight measurements, which is insufficient to substantiate the effects of Prosopis laevigata

2)The MIC experiment does not provide the actual minimum inhibitory concentration (MIC) value. Simply stating 'less than 25' is meaningless without precise data

3)The discussion section barely addresses the inflammatory response, with only a brief mention in the final paragraph, despite the fact that the paper is centered on anti-inflammatory effects

4)The materials and methods, results, discussion, as well as the abstract, are not up to the mark, and the English language needs improvement. Lines 23 to 25 are especially confusing, as they suggest you are testing the solvent on the mice

Comments on the Quality of English Language

It needs improvement. 

Author Response

The manuscript has many flaws that cannot be corrected without repeating the experiments. Some of them are as follows:

1)The manuscript claims an anti-inflammatory effect starting from the title, but the authors base this conclusion solely on weight measurements, which is insufficient to substantiate the effects of Prosopis laevigata

Response: Thank you for your recommendation. TPA is an irritant isolated from Croton tiglium L. that causes skin edema, when applied to the ear of mice it causes erythema, the ear thickens and there is extravasation of fluid and redness within the first hours after application. The edema caused by this substance is part of the inflammatory process, in this first paper on the study of Prosopis laevigata, the objective was to perform a chemical characterization based on two assays, the antimicrobial one and the local edema assay caused by TPA because it is recognized that this substance causes all processes associated with inflammation and that the thickness of the ear is the first variable analyzed as a measure of that event, and because this is the first pharmacological approach made of this Mexican species and that we observed a significant inhibition of edema, an inflammation study is being planned to evaluate its chronic effect and thus be able to measure associated biochemical parameters.

2.- The MIC experiment does not provide the actual minimum inhibitory concentration (MIC) value. Simply stating 'less than 25' is meaningless without precise data

Response: Your recommendation is understandable and we are proposing to carry out this study at a value where the minimum inhibitory concentration is used in subsequent studies. At the moment, the extract shows an effect as an antimicrobial at the concentrations evaluated, the lowest of which was 25 mg/mL. This first result directs us towards the active metabolites.

3.- The discussion section barely addresses the inflammatory response, with only a brief mention in the final paragraph, despite the fact that the paper is centered on anti-inflammatory effects

Response: Thank you very much for your comment. A paragraph was added to explain more about compounds with anti-inflammatory activity.

4.- The materials and methods, results, discussion, as well as the abstract, are not up to the mark, and the English language needs improvement. Lines 23 to 25 are especially confusing, as they suggest you are testing the solvent on the mice

Response: The document was sent to an English language expert. The document was carefully reviewed.

Round 2

Reviewer 1 Report

Comments and Suggestions for Authors

I have gone through the revised comments. 

The authors did not perform any recommended experiments that are important to improve the quality of the manuscript even though the experiments do not require longer to do. The author's response is unsatisfactory. 

Author Response

The authors did not perform any recommended experiments that are important to improve the quality of the manuscript even though the experiments do not require longer to do. The author's response is unsatisfactory. 

Response: We appreciate your comment, but the experiment to determine the cytokines will be carried out using another anti-inflammatory model in future research because it involves registering and using additional animals for the project, which is why authorization from the Ethics Committee is required, and the resources for the development and implementation of this methodology are limited.

Reviewer 2 Report

Comments and Suggestions for Authors

Dear authors,

The manuscript's current version is definitely better than the previous version. However, it is a great pity that the text still contains many errors and basic inconsistencies (even those that were previously commented on..). For this reason, I think the text needs to be reviewed again and in detail to unify and correct errors and inconsistencies (I will not write everything again).

Examples:

1. Chapter 4.9.1 - many names of microorganisms are incorrectly taxonomically listed! See "epidermis"!!

2. inconsistent and unclear designations are often used - see Staphylococcus, Staphylococcus, S. ....... (it makes no sense). The same applies to other MO, E. coli, .....

3. The designation "Gram-positive" (capital G). etc. is more appropriate.

4. L357 - 108 or exponent?? See also my previous comment!

Author Response

The manuscript's current version is definitely better than the previous version. However, it is a great pity that the text still contains many errors and basic inconsistencies (even those that were previously commented on..). For this reason, I think the text needs to be reviewed again and in detail to unify and correct errors and inconsistencies (I will not write everything again).

Examples:

  1. Chapter 4.9.1 - many names of microorganisms are incorrectly taxonomically listed! See "epidermis"!!
  2. inconsistent and unclear designations are often used - see Staphylococcus, Staphylococcus, S. ....... (it makes no sense). The same applies to other MO, E. coli, .....
  3. The designation "Gram-positive" (capital G). etc. is more appropriate.
  4. L357 - 108 or exponent?? See also my previous comment

Response: We appreciate your comments, which helped make the document complete with each of your observations. The document was carefully revised in the points in chapter 4.9.1 and the taxonomy of microorganisms was corrected, as well as the spelling of the same. In line 357, exponent 108 was corrected.

Round 3

Reviewer 2 Report

Comments and Suggestions for Authors

I am satisfied with the revised manuscript. 

Author Response

Response: We appreciate your comments, which helped make the document complete with each of your observations. The document was carefully revised in the points in chapter 4.9.1 and the taxonomy of microorganisms was corrected, as well as the spelling of the same. In line 357, exponent 108 was corrected.